# Comparison of Antibiotic Resistance Profile of *Escherichia coli* between Pristine and Human-Impacted Sites in a River

**DOI:** 10.3390/antibiotics10050575

**Published:** 2021-05-13

**Authors:** Emi Nishimura, Masateru Nishiyama, Kei Nukazawa, Yoshihiro Suzuki

**Affiliations:** 1Department of Civil and Environmental Engineering, Faculty of Engineering, University of Miyazaki, Miyazaki 889-2192, Japan; 252524mura@gmail.com (E.N.); nukazawa.kei.b3@cc.miyazaki-u.ac.jp (K.N.); 2Department of Food, Life and Environmental Science, Faculty of Agriculture, Yamagata University, Tsuruoka 997-8555, Japan; m-nishiyama@tds1.tr.yamagata-u.ac.jp

**Keywords:** antibiotic-resistant *E. coli*, river, pristine river, multidrug resistance, PFGE similarity

## Abstract

Information on the actual existence of antibiotic-resistant bacteria in rivers where sewage, urban wastewater, and livestock wastewater do not load is essential to prevent the spread of antibiotic-resistant bacteria in water environments. This study compared the antibiotic resistance profile of *Escherichia coli* upstream and downstream of human habitation. The survey was conducted in the summer, winter, and spring seasons. Resistance to one or more antibiotics at upstream and downstream sites was on average 18% and 20%, respectively, and no significant difference was observed between the survey sites. The resistance rates at the upstream site (total of 98 isolated strains) to each antibiotic were cefazolin 17%, tetracycline 12%, and ampicillin 8%, in descending order. Conversely, for the downstream site (total of 89 isolated strains), the rates were ampicillin 16%, cefazolin 16%, and tetracycline 1% in descending order. The resistance rate of tetracycline in the downstream site was significantly lower than that of the upstream site. Furthermore, phylogenetic analysis revealed that many strains showed different resistance profiles even in the same cluster of the Pulsed-Field Gel Electrophoresis (PFGE) pattern. Moreover, the resistance profiles differed in the same cluster of the upstream and the downstream sites. In flowing from the upstream to the downstream site, it is plausible that *E. coli* transmitted or lacked the antibiotic resistance gene.

## 1. Introduction

In medical institutions, antibiotic-resistant bacteria (ARB) have emerged, and nosocomial infections caused by ARB have become a worldwide problem. The number of deaths from ARB globally is reported to be 700,000 annually, and it is estimated that the number of deaths will increase to 10 million by 2050 [1]. In Japan, the number of deaths due to ARB is not clear. However, 8000 deaths were confirmed due to two typical ARB bloodstream infections in 2017 [2]. Therefore, countermeasures against the challenges posed by ARB are being promoted by national and international organizations. The World Health Organization and the Center for Disease Control and Prevention (CDC) have showed that ARB pose a serious threat to the world and published survey data to warn of the severity of the problem [3,4]. Despite this, the spread of ARB does not seem to have been suppressed. ARB have been detected not only in medical institutions, livestock farms and aquaculture farms that frequently use antibiotics, but also in living areas such as food, treated sewage, and urban rivers [5,6]. As of 2021, many research results on the emergence and spread of ARB and multidrug-resistant bacteria resistant to various antibiotics in the human living sphere are being reported regularly. Furthermore, ARB have been recently detected in wild animals and natural environments that are not directly affected by humans, and this is a cause for concern. For example, in Portugal, 28 of the 218 *Escherichia coli* strains isolated from wild birds’ feces flying to distant islands showed antibiotic resistance [7]. In the United States, it has been reported that 20 of 22 types of bacteria isolated from wild killer whales inhabiting the ocean near the coast and its surroundings contain antibiotic resistance [8]. From two marine mammals (*Phoca vitulina* and *Porpoise phocoena*) along the coast of Washington, USA, 37% (out of 144 strains) of isolated strains were resistant to at least one drug, and 26% showed multidrug resistance [9]. In Mexico, chloramphenicol-resistant strains have been detected in the surface waters of isolated cave microbiome [10]. A study of polar snow and glaciers (where humans have not intervened) revealed the presence of resistance genes due to clinical (i.e., *aac (3)* and *bla_IMP_*) and agricultural (i.e., *strA* and *tetW*) tests. These were spread through bacteria via air, and migratory birds have been pointed out as their infection routes [11].

In this way, circumstantial evidence of ARB spread throughout the globe continues to be reported. Therefore, it is vital to provide information on ARB’s existence in natural environments that are not directly exposed to antibiotics, especially rivers. However, information on ARB’s distribution and profile in pristine streams, as well as changes in antibiotic resistance due to the flow process, is still largely lacking today. Even in mountain streams near water sources surrounded by forests, contamination by feces such as that of wild animals will be possible, and antibiotic-resistant *E. coli* (AR-*E. coli*) may be detected. This study targeted the Kaeda River, a river whose source is separated from human activity by a natural forest and examined the actual existence of AR-*E. coli* at both the upstream site in the deep forest and the downstream site where human habitations are formed (Figure 1). Antibiotic susceptibility tests for various antibiotics were conducted on the strains collected and identified from each site. After that, the antibiotic resistance profile was created, and the genotypes were analyzed using pulsed-field gel electrophoresis (PFGE). We examined the possibility of resistance profiles changing with the flow of the river from the similarity between the antibiotic resistance profile and the genotype.

## 2. Results and Discussion

### 2.1. Water Qualities in the Kaeda River

Water quality analyses and bacterial counts at each survey site are shown in Table 1. The water temperatures at the pristine upstream and human-impacted downstream sites were highest in July (summer), at 21.2 °C and 28.3 °C, respectively. The temperature remained around 12 °C in December (winter), and April (spring), and the water temperature difference between upstream and downstream was less than 2 °C. The pH ranged from 6.3 to 6.7, slightly lower than neutral. The electric conductivity (EC) differed between the upstream and downstream sites, averaging 79 μS/cm and 91 μS/cm, respectively. It was considered that the mineral component increases due to the flow process. The turbidity was extremely low on all survey days, with mean values of 0.57 and 0.86 degrees of turbidity units, and the sample water was extremely clear. The dissolved oxygen (DO) was near the saturation concentration at each water temperature, except for the downstream site in July. The DO supersaturation at the downstream site in July was thought to be due to photosynthesis by periphyton algae. The total organic carbon (TOC) concentration was low and ranged from 0.34 to 0.58 mg C/L in all survey data, though it was noted to be higher in the downstream sites. The treated domestic wastewater from the human habitation area to the downstream loaded a small number of organic substances.

The numbers of *E coli* and enterococci were constantly detected at the upstream site and were 2 to 1.8 × 10^1^ colony-forming units (CFU)/100 mL and 5.0 to 3.6 × 10^1^ CFU/100 mL, respectively. It was found that fecal bacteria are constantly loaded upstream, which is located in the deep forest. Due to seasonal changes in the numbers of *E. coli* and enterococci at the upstream site, the number of bacteria in July was one order higher than that in December. In July, many birds were observed in the upstream area, and birds and weasels’ feces were observed on the riverside. Increased bacterial numbers in the upstream site may be associated with wildlife activity due to changes in temperature conditions. Comparing the results of the three surveys at upstream and downstream sites, numbers of *E. coli* and enterococci were higher at the downstream site. The downstream site of the Kaeda River was loaded with fecal bacteria from the human habitation area.

### 2.2. Identification of E. coli Isolates from the Kaeda River

Of all 360 strains (pristine upstream, 180 strains; human-impacted downstream, 180 strains) that were determined to be positive colonies by CHROMagar ECC agar plate, 51.1% (184/360 strains) were identified as *E. coli* by PCR analysis targeting *uspA*. The average upstream and downstream site identification rates in the three surveys were 52.7% and 49.3%, respectively. Among the positive colony strains, there were pseudo positives or *E. coli* strains that did not retain *uspA*. When studying *E. coli* isolated from the natural environment, it is necessary to identify the isolated strains meticulously. In this study, strains confirmed to have *uspA* were defined as *E. coli*-identified strains and used for antimicrobial susceptibility testing and PFGE analysis.

### 2.3. Seasonal Changes in the Antibiotic-Resistant Rate of E. coli

The antimicrobial susceptibility test was conducted on 98 and 89 strains of *E. coli*-identified from the upstream and downstream sites, respectively. Table 2 shows the number of AR-*E. coli* strains and the resistance rate during the survey period. During the survey, 18% (18/98 strains) of the upstream site and 23% (20/89 strains) of the downstream site strains showed resistance to one or more antimicrobials. In July, the antimicrobial resistance rate was the highest in both the upstream and downstream sites, and was 45% (15/33 strains) and 31% (12/39 strains), respectively. Conversely, in December, the antibiotic resistance rate decreased significantly in both the upstream and downstream sites, reaching 4% (1/24 strain) and 5% (2/41 strain), respectively. No association was found between the change in resistance rate and the increase or decrease in the number of *E. coli* during the survey. Seasonal fluctuations in the detection rate of AR-*E. coli* were confirmed at both sites.

In the Kaeda River, which is located in a very natural area of Japan, it is fascinating that the antibiotic resistance rate of *E. coli* at the upstream site was 46% in July when the number of bacteria was also high. The AR-*E. coli* was detected in the range of 0 to 2.5% in mountain torrent water with relatively little human pollution near Tokyo, a Japanese mega-city [12]. The resistance rate of *E. coli* at the pristine upstream site of the Kaeda River was therefore 12 times higher than that of the stream near Tokyo. It was suggested that the resistant strains’ rate did not directly reflect the effects of the human living sphere. In July, when the river’s water temperature is high, the AR-*E. coli* spreads to the upstream site of the river where the catchment area is covered with forest. It has been reported that wild animals can become carriers that acquire ARB in the human living sphere and spread it to various environments [7,13]. A study of AR-*E. coli* for American seagulls reported that AR-*E. coli* increased in densely populated areas across vast continental regions such as Alaska, Russia, and Canada [14]. As a result, it is considered that the antibiotic resistance rate of *E. coli* increases in July when wild animals, such as birds, load AR-*E. coli* to the upstream site. Elucidation of the factors that control the antibiotic resistance rate of bacteria, the sources of ARB contamination, and their routes in pristine streams, will be fundamental issues in the future.

### 2.4. Comparisons of Antibiotic Resistance Rate and Resistance Patterns

Figure 2 shows the percentages of one to three antibiotics-resistant strains to all strains collected from the pristine upstream and human-impacted downstream sites. The one antibiotic-resistant strains were 3% (3/98 strains) and 12% (11/89 strains), respectively. The two antibiotics-resistant strains of the upstream and downstream sites were 11.2% (11/98 strains) and 9% (8/89 strains). The three antibiotics-resistant strains that became multidrug-resistant were higher in the upstream site at 4% (4/98 strains) than in the downstream site at 1% (1/89 strains). The antibiotic resistance rates of *E. coli* to each antibiotic, MIC50 and MIC90, are summarized in Table 3. The AR-*E. coli* resistance to ampicillin (ABPC), cefazolin (CEZ), and tetracycline (TC) were detected from both the upstream and downstream sites. In the upstream site, the resistance rate of cefazolin was highest (17.3%, 17/98 strains), followed by ABPC (8%, 8/98 strains) and TC (12%, 12/98 strains). The MIC90 of cefazolin was 32 μg/mL, and 10% of *E. coli* strains showed a resistance to cefazolin at the upstream site. In contrast, the resistance rates of ABPC (17%, 15/89 strains) and cefazolin (18%, 16/89 strains) were high in the downstream site. However, the resistance rate of TC was significantly lower (1%, 1/89 strain) than that of the upstream site.

The MIC90 of ABPC for the downstream site was 64 μg/mL, which was twice as high as the tolerance criterion of the Clinical and Laboratory Standards Institute (CLSI). From the upstream and downstream sites, *E. coli* resistant to the β-lactam antibiotics ABPC and CEZ, which are important in treating human *E. coli* infections, were detected in the range of 8 to 17%. The resistance rates to ABPC and CEZ are 53% and 39%, respectively, higher than the resistance rates of other β-lactam antibiotics [15]. TC, which showed a high resistance rate in the upstream site, is frequently used in the medical and livestock fields. TC antibiotics are used worldwide for poultry, and *E. coli* has been reported to have a resistance rate of 87% (out of 2164 strains), especially in China [16]. In Japan, the amount of TC antibiotics used is highest for animals, and the resistance rate isolated from pigs exceeds 50% [17]. The fact that strains resistant to the above antibiotics used in humans and livestock have been detected from the upstream forest site through all surveys suggests that the TC-resistant *E. coli* exists in the natural water environment regardless of the season. Although the transportation route to the upstream site is not clear, it is highly likely that ARB originating from humans and livestock may have spread to the upstream area in the forest.

Next, the antibiotic resistance of the strains was profiled, and the patterns of antibiotic resistance profiles were compared for the upstream and downstream sites based on the results obtained in the antibacterial susceptibility test (Table 4). In July, four patterns of resistance profiles were confirmed from the upstream and downstream sites, respectively. In common between upstream and downstream sites, resistant strains with the ABPC-CEZ and CEZ-TC patterns were detected. Moreover, 15% of resistant strains of the ABPC alone were detected only at the downstream site in July. In December (winter), the resistance rate was extremely low, and each pattern was independent. In April (spring), resistant strains of the ABPC-CEZ pattern were detected from both the upstream and downstream sites. In two surveys in July and April, the same resistance pattern was confirmed in the upstream and downstream sites, suggesting transportation of resistant strains from the upstream to downstream sites. Therefore, we further followed up genotyping by PFGE analysis.

### 2.5. Comparison of Antibiotic Resistance Profiles and PFGE Types

The PFGE genotyping was analyzed for 72 strains (upstream, 33 strains; downstream, 39 strains) in July, in which AR-*E. coli* was most frequently detected. As a result, 68 out of 72 *E. coli* strains were genotyped, and 4 strains were excluded from the analysis because no clear genotype could be obtained, even after multiple analyses. A phylogenetic analysis was performed using the acquired PFGE types and compared with the antibiotic resistance profile (Figure 3). The PFGE type was classified into 40 types, with a similarity of 1.0. The PFGE type of *E. coli* isolated from rivers was highly diverse. In addition, the phylogenetic tree classified the strains into 12 clusters. Since the strains in the cluster are of the same PFGE type, they can be regarded as identical clones. Many strains showed resistance to different antibiotics even in the same cluster, suggesting that *E. coli* strains retained different antibiotic resistance genes, even though they are the same clone.

Furthermore, the strains isolated from the upstream and downstream sites were classified into the same cluster (C-type, I-type, K-type). It was found that *E. coli* loaded to the river at the upstream site was transported to the downstream site. In this case, it should also be noted that the antibiotic resistance profiles were different. For example, in the C-type cluster, the upstream strain was a strain resistant to ABPC, cefazolin, and TC (multidrug-resistant strain), while the downstream strain was resistant to cefazolin and TC. In the flowing process from the upstream to downstream sites, it is possible that *E. coli* transmitted or lacked antibiotic resistance. Although as different species of bacteria, studies on enterococci have shown results suggesting the acquisition or deficiency of resistance genes in river water [18]. Antibiotic resistance profiles related to retention of the resistance genes based on genome sequencing analysis for *E. coli* strains were isolated from the upstream site of the Kaeda River [19]. At present, there have been no cases of tracking the acquisition or deletion of antibiotic resistance genes in the process of the river flowing downstream, but the possibility cannot be ruled out. We have confirmed that antibiotic resistance developed in natural *E. coli* strains after mixing with sewage-treated effluent in the river [20].

## 3. Materials and Methods

### 3.1. Sampling

Samples of river water were collected from the Kaeda River (length of river channel: 17.5 km; catchment area: 53.8 km^2^) (Figure 1) which flows through the Miyazaki city in the Miyazaki prefecture, Japan. The catchment area of the Kaeda River is the forest around the riverside from downstream sites. Since the upstream site in Kaeda River is located in the deep forest, there is no load of artificial human activities. Since the wasteland near the upstream site is deforested land due to typhoon events, there is no human activity. The downstream site receives the effluent discharged from the small community (5000 people, no sewage system). Sampling of the Kaeda River was conducted on July 29, 2016 (summer), December 26, 2016 (winter), and April 6, 2017 (spring). The samples were taken from each site between 10:00 and 12:00 h. There was no rainfall on the day of sampling, or for a couple of days before. Samples were immediately transported to the laboratory, and the enumeration of bacteria and water quality analyses was done within 4 h after sample collection. Water temperature and dissolved oxygen were determined on the site by a fluorescent-type dissolved oxygen meter (HQ40d, HACH Co., Loveland, CO, USA). A benchtop pH/water quality analyzer (LAQUA, Horiba, Kyoto, Japan) was used to measure pH and EC. Turbidity was determined using a turbidity meter (SEP-PT-706D; Mitsubishi Kagaku, Tokyo, Japan). The concentrations of TOC in the samples were determined using a TOC analyzer (TOC-V Model, Simadzu Co., Kyoto, Japan).

### 3.2. Enumeration of Fecal Indicator Bacteria

Each water sample (upstream water, 100–2000 mL; downstream water, 10–100 mL) was filtered through a sterile 0.45 μm-pore membrane filter (47 mm diameter, mixed cellulose ester; Advantec, Tokyo, Japan) to capture microbial cells. Membranes were placed on CHROMagar ECC agar plates (CHROMagar, France) and incubated at 37 °C for 24 h. Blue colonies were considered to be *E. coli*. Enterococci were enumerated using the membrane filter method with membrane–Enterococcus indoxyl-β-D-glucoside agar (mEI) plates [21]. The samples passed through a membrane filter incubated on mEI agar plates at 41 °C for 24 h. After incubation, colonies on the filter that had blue halos were regarded as enterococci. Concentrations of *E. coli* and enterococci in each sample were expressed as mean CFUs per 100 mL of three replicates. The detection limit of the method of this analysis was 0.3 CFU/100 mL.

### 3.3. Identification of E. coli by PCR Analysis

Presumptive *E. coli* were streaked on Brain–Heart Infusion agar (BHI, Difco Laboratories, Detroit, MI, USA) and were incubated overnight at 37 °C. After incubation, a single colony was suspended in 100 µL of sterile distilled water as template DNA for PCR analysis. The primers used for *uspA* amplification were shown to be gene-specific for *E. coli* [22]. The enzyme used for PCR amplification was KAPA Taq Extra (Kapa Biosystems, Nippon Genetics, Tokyo, Japan). The reaction mixture (15 µL) contained 1× KAPA Extra Buffer, 2.5 mM MgCl_2_, 0.3 mM dNTPs, 0.5 µM each of forward and reverse primers, 0.1 U KAPA Taq Extra DNA polymerase, and 1.0 µL of template DNA. The PCR amplification program used was as follows: initial denaturation at 94 °C for 5 min, 30 cycles of denaturation at 94 °C for 2 min, annealing at 70 °C for 1 min, elongation at 72 °C for 1 min. Amplification was confirmed by electrophoretic analysis of a 5 µL aliquot of the reaction product, mixed with 1 µL of 6× Loading Buffer (TAKARA, Tokyo, Japan) on 1.0% agarose gel. The strain NBRC 3301 (*E coli*; National Institute of Technology and Evaluation, Japan) was used as a positive control for identifying *E. coli* by PCR analysis.

### 3.4. Determination of Minimum Inhibitory Concentration (MIC)

The MIC of each antibiotic was determined using the agar dilution method according to the CLSI guidelines [23,24]. According to the recommendations of the CLSI, the 9 kinds of antibiotics for the Gram-negative bacteria were tested as follows: ampicillin (ABPC, Wako Pure Chemical Industries, Osaka, Japan), cefazolin (CEZ, Wako Pure Chemical Industries), cefotaxime (CTX, Wako Pure Chemical Industries), imipenem (IPM, Wako Pure Chemical Industries), gentamicin (GM, Wako Pure Chemical Industries), tobramycin (TOB, Wako Pure Chemical Industries), ciprofloxacin (CPFX, Wako Pure Chemical Industries), chloramphenicol (CP, Sigma-Aldrich, St. Louis, MO, USA), tetracycline (TC, Wako Pure Chemical Industries). All antibiotics were dissolved and diluted according to CLSI guidelines and antimicrobial solutions were used on the day of preparation. The plates contained two-fold dilutions of antibiotics with ten-grade concentration ranging as follows: 0.25 to 128 μg/mL for ABPC and CP; 0.0625 to 32 μg/mL for CEZ; 0.03 to 16 μg/mL for CTX, IPM and CPFX; 0.125 to 64 μg/mL for GM, TOB and TC. The identified E. coli strains were cultured for 18 h in Mueller Hinton broth (MH, Becton, Dickinson and Company, Franklin Lakes, NJ, USA). Inocula were then applied to the surface of MH agar (1.7% agar) plates containing various concentrations of each antibiotic using Micro Planter (Sakuma Co., Tokyo, Japan). The plates were incubated at 37 °C for 18 h, and MICs were determined. MIC breakpoints for intermediate and resistant samples were based on CLSI criteria [24]. A reference strain of E. coli ATCC25922 was used as a quality control.

### 3.5. PFGE Typing

PFGE was performed according to the standard PFGE protocol for *E. coli* genotyping [25]. In brief, freshly grown *E. coli* cells on BHI agar were transferred to 1 mL of cell suspension buffer (TE buffer; 100 mM Tris, 100 mM EDTA, pH 8.0) by using a cotton swab. An aliquot (100 µL) of the cell suspension was transferred to a new microcentrifuge tube and treated with 5.0 µL of Proteinase K (20 mg/mL) at 60 °C for 3 min. The treated cell suspension was mixed with 100 µL of melted 2% CleanCut agarose (Bio-Rad), and transferred to disposable plug molds (Bio-Rad). Once solidified, gel plugs were transferred to a 15 mL polypropylene screwcap tube and incubated with 25 µL of proteinase K (20 mg/mL) in 5 mL of cell lysis buffer (50 mM Tris; 50 mM EDTA, pH 8.0 + 1% sarcosyl) at 55 °C for 4 h. After incubation, the plugs were washed five times with TE buffer at 50 °C for 10 min. Genomic DNA was digested with 50 U of Xba I (Takara Bio). PFGE was performed at 14 °C for 20 h on 1% pulsed-field certified agarose (Bio-Rad) in 0.5× Tris-Borate-EDTA (TBE) by using a CHEF-DR II system (Bio-Rad). Pulse times were increased from 6.8 s to 35.4 s during a 20 h run at 6 V/cm. Lambda DNA ladders with a size range of 48.5–873 kb (Lonza) were used as a size marker.

Analysis of band-based PFGE patterns was performed using a gene profiler software (Scanalytics, Buckinghamshire, UK). Levels of similarity between fingerprints were expressed as Dice coefficients. PFGE patterns were clustered using the unweighted pair group method with arithmetic means. PFGE patterns similar to each other at the 1.0 similarity level (=100% similarity) were considered identical.

## 4. Conclusions

We compared the existence and characteristics of AR-*E. coli* at the upstream site in the natural forest and the downstream site in the human habitation area. The antimicrobial resistance rate of *E. coli* was extremely high in the summer, at 46% (15/33 strains) and 31% (12/39 strains) in the upstream and downstream, respectively. These strains were resistant to ABPC, CEZ, and TC. Several AR-*E. coli* were detected at the upstream site of the forest far from the active human sphere. However, in winter, the resistance rate decreased significantly at both sites and reached 4% (1/24 strain) at the upstream site and 5% (1/19 strain) at the downstream site. The antibiotic resistance rate of *E. coli* in rivers fluctuates greatly depending on the season. Presently, however, AR-*E. coli*’s governing factors are still unknown. Questions on the source of AR-*E. coli* in the river upstream and the transport media of AR-*E. coli* to the upstream are yet to be fully answered. The relationship between the animal ecosystem of the entire basin and the human living sphere is important. Moreover, PFGE analysis revealed that *E. coli* loaded on the river’s upstream site is transported downstream. However, the antibiotic resistance profiles were different even in the same cluster of the PFGE type, suggesting that the antibiotic resistance gene may have been transmitted or deleted from *E. coli* in the flowing process from the upstream to downstream sites. Elucidation of the actual source and pollution route of ARB in the aquatic environment is the most critical issue for improving public health.

## Figures and Tables

**Figure 1 antibiotics-10-00575-f001:**
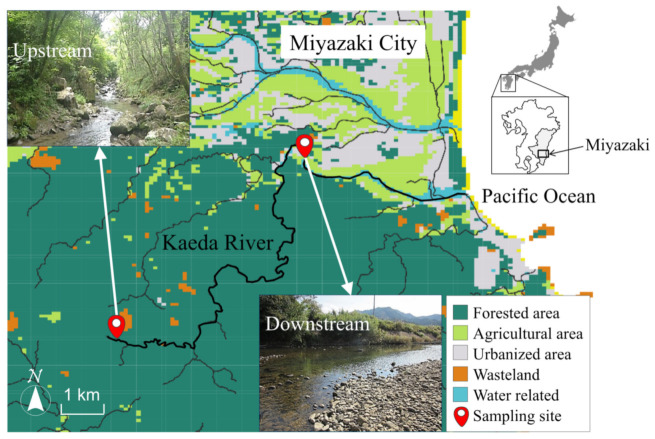
Sampling sites of pristine upstream and human-impacted downstream at the Kaeda River in southern Japan, with distribution of major land use classifications.

**Figure 2 antibiotics-10-00575-f002:**
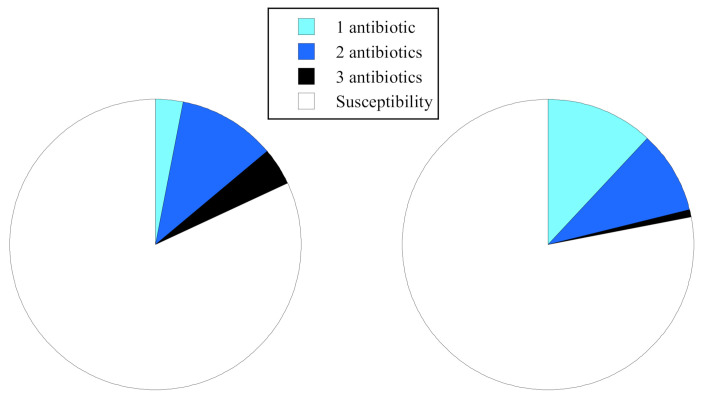
Percentages of one to three antibiotics-resistant strains to each site collected from the pristine upstream and human-impacted downstream sites.

**Figure 3 antibiotics-10-00575-f003:**
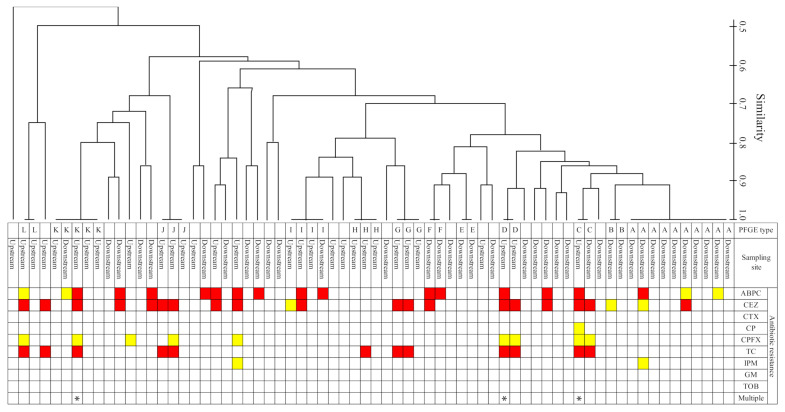
Dendrograms of pulsed-field gel electrophoresis types and antibiotic resistance profiles for *E. coli* isolates from each sampling site in the Kaeda River. Concerning antibiotic resistance phenotypes: red indicates resistance, yellow indicates intermediate resistance, and white indicates susceptibility. Ampicillin, ABPC; cefazolin, CEZ; cefotaxime CTX; chloramphenicol, CP; ciprofloxacin, CPFX; tetracycline, TC; imipenem, IPM; gentamicin, GM; tobramycin, TOB.

**Table 1 antibiotics-10-00575-t001:** Water qualities and bacterial counts at each site during survey period.

Parameter	Pristine Upstream			Human-Impacted Downstream		
	29 Jul., 2016	26 Dec., 2016	6 Apr., 2017	29 Jul., 2016	26 Dec., 2016	6 Apr., 2017
Water temp (°C)	21.2	11.2	12.6	28.3	12.4	14.2
DO (mg/L)	8.4	9.9	9.5	9.4	10	9.7
pH (-)	6.4	6.5	6.3	6.7	6.6	6.5
EC (μS/cm)	62	110	60	93	98	81
Turbidity (kaolin unit)	1.1	0.12	0.42	1.2	0.28	1.0
TOC (mg-C/L)	0.40	0.37	0.34	0.58	0.57	0.46
Total coliform (CFU/100 mL)	(7.8 ± 1.4) * × 10^2^	(2.2 ± 0.5) × 10^2^	(3.0 ± 0.2) × 10^2^	(1.6 ± 0.3) × 10^3^	(7.2 ± 0.9) × 10^2^	(8.0 ± 0.4) × 10^2^
*Escherichia coli* (CFU/100 mL)	(1.8 ± 0.2) × 10^1^	2 ± 2	(1.3 ± 0.1) × 10^1^	(3.7 ± 1.7) × 10^1^	(8.0 ± 1.6) × 10^1^	(4.3 ± 1.9) × 10^1^
Enterococci (CFU/100 mL)	(3.6 ± 1.7) × 10^1^	5 ± 0.1	5 ± 0.3	(5.3 ± 0.4) × 10^1^	(1.5 ± 0.3) × 10^1^	7 ± 4

* mean ± SD (standard deviation), *n* = 3.

**Table 2 antibiotics-10-00575-t002:** Antibiotic resistance rate of *E. coli* during survey period.

Sampling Point	Pristine Upstream (98 Isolates)				Human-Impacted Downstream (89 Isolates)			
Date	29 Jul., 2016	26 Dec., 2016	6 Apr., 2017	Total	29 Jul., 2016	26 Dec., 2016	6 Apr., 2017	Total
Resistance rate (%)	46	4	5	18	31	5	23	23
(Isolates)	(15/33)	(1/22)	(2/41)	(18/98)	(12/39)	(1/19)	(7/31)	(20/89)

**Table 3 antibiotics-10-00575-t003:** Antibiotic susceptibility of *E. coli* isolates from Kaeda River.

Group	Antimicrobial Agent	MIC Test Range (μg/mL)	Pristine Upstream (98 Isolates)					Human-Impacted Downstream (89 Isolates)				
			Susceptible	Intermediate	Rsistant	MIC50 (μg/mL)	MIC90 (μg/mL)	Susceptible	Intermediate	Rsistant	MIC50 (μg/mL)	MIC90 (μg/mL)
			No. Isolates (% Isolates)					No. Isolates (% Isolates)				
Penicillins	ABPC	0.25–128	89 (91%)	1 (1%)	8 (8%)	2	8	69 (78%)	5 (6%)	15 (16%)	2	64
Cephem	CEZ	0.0625–32	67 (68%)	14 (14%)	17 (17%)	1	32	58 (65%)	17 (19%)	14 (16%)	1	16
	CTX	0.03–16	98 (100%)	0 (0%)	0 (0%)	0.03	0.25	89 (100%)	0 (%)	0 (%)	0.03	0.125
Aminoglycosides	GM	0.125–64	98 (100%)	0 (0%)	0 (0%)	1	2	89 (100%)	0 (%)	0 (%)	0.5	2
	TOB	0.125–64	98 (100%)	0 (0%)	0 (0%)	0.5	2	89 (100%)	0 (%)	0 (%)	0.5	2
Carbapenems	IPM	0.03–16	97 (99%)	1 (1%)	0 (0%)	0.06	0.5	86 (97%)	3 (3%)	0 (%)	0.06	0.25
Fluoroquinolons	CPFX	0.03–16	98 (100%)	0 (0%)	0 (0%)	—	0.03	86 (97%)	3 (3%)	0 (%)	—	0.03
Tetracyclines	TC	0.125–64	86 (88%)	0 (0%)	12 (12%)	2	8	88 (99%)	0 (%)	1 (1%)	4	8
Chloramphenicols	CP	0.25–128	96 (98%)	2 (2%)	0 (0%)	1	32	89 (100%)	0 (%)	0 (%)	1	4

**Table 4 antibiotics-10-00575-t004:** Patterns of antibiotic resistance profiles.

Year	Sampling Date	Sampling Point (*n* = Isolates)	Resistance Profile	Number of Isolates (%)
2016	29 Jul.	Upstream	Susceptibility	18 (55%)
	(Summer)	(*n* = 33)	TC	1 (3%)
			ABPC-CEZ	3 (9%)
			CEZ-TC	7 (21%)
			ABPC-CEZ-TC	4 (12%)
		Downstream	Susceptibility	27 (69%)
		(*n* = 39)	ABPC	6 (15%)
			CEZ	1 (4%)
			ABPC-CEZ	4 (10%)
			CEZ-TC	1 (3%)
2016	26 Dec.	Upstream	Susceptibility	23 (96%)
	(Winter)	(*n* = 24)	ABPC	1 (4%)
		Downstream	Not examined	18 (95%)
		(*n* = 19)	CEZ	1 (5%)
2017	6 Apr.	Upstream	Susceptibility	39 (95%)
	(Spring)	(*n* = 41)	CEZ	1 (2%)
			ABPC-CEZ	1 (2%)
		Down stream	Susceptibility	24 (77%)
		(*n* = 31)	ABPC	1 (3%)
			CEZ	3 (10%)
			ABPC-CEZ	2 (7%)
			ABPC-CEZ-TC	1 (3%)

## Data Availability

Not applicable.

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
