# Peer review of "Comparison of Antibiotic Resistance Profile of Escherichia coli between Pristine and Human-Impacted Sites in a River"

_antibiotics, 2021, doi:10.3390/antibiotics10050575_

Round 1
Reviewer 1 Report
In this study the presence of antimicrobial resistant E. coil up- and downstream of human habitation is investigated. Overall there is no significant difference between rates of resistance up- verses downstream, although interestingly there is more resistance to cephems upstream. There are changes in rates of resistance by season, with lower rates in winter.
This is a well conducted study and a well written paper. Results are clearly and concisely presented and I have no hesitation is recommending it for publication subject to a few minor points. Reporting of the statistics used is the most vital correction needed.
General points:
1. Generally writing is clear and concise, and everything is understandable. However I have noted a few errors and a general read-though to catch any errors would be worthwhile.
2. I am not keen on the title, in particular the use of "natural river". All rivers are natural, otherwise they would be a canal. Please drop the use of "natural river" thought-out the paper. The title (and study) should reflect that is it upstream and downstream of human settlement/activity. Therefore you have pristine river upstream, and human impacted river downstream.
2.1 Related, but a better definition of the difference between the sites, (e.g. human impacted or whatever) needs to be included in the text, e.g. line 13 but also throughout.
3. Statistical methods are not reported in the methods, but there are some P-values. It is not clear how statistics where calculated, as there is just one set of samples for each location and each time of year, so n=1. This needs to be cleared up before publication. Also, when reporting stats include degrees freedom, test-statistic, and effect size/coefficient.
3.1 on a related note, be carful saying something is "significant" when no statistical test was carried out, e.g. remove "significant" lines 123, 161, and elsewhere, unless a statistical test was carried out.
4. Due to the journal putting the methods at the end, figure 1 (map) is out of place and needs to be moved or renamed.
4.1 on the map, it would be good to put a pin showing the location of Miyazaki.
Specific points:
Abstract.
Line 12-13, could be more simple and clear. e.g. This study compared the antibiotic resistance profile of Escherichia coli upstream and downstream of human habitation.
Line 15, antibiotics cant have antibiotic resistance. I think you mean
Resistance to one or more antibiotics...….
Introduction:
Line 38, "selected" is the wrong word.
Line 44, reported daily? is there evidence of a paper per day? Regularly might be a better word.
Line 66, "away from the humans" is clumsy. Away from human activity would be better.
Results:
Table 1. are these values means? what is the standard deviation, how did you did statistics if no standard deviation or variance? please put the standard deviation on the plots. Would be even better to put a star or something to indicate values that are significantly different up- and downstream. (remember to detail stats in methods).
Table 2. again you have stats, but not clear how as you don't have standard deviation?
Line 148, be consistent with numbers "one to three" or "1 to 3",
Table 3, is cephemes spelt right? I think it is cephem?
Methods:
Please provide more detail for the MIC testing. Unfortunately there is great variation in how the CLSI guidelines are interpreted, especially for environmental samples. Make it clear exactly what concentrations where used for each antibiotic, rather than the range (e.g. ampicillin at x, y, and z ug/mL, cegazolin at a, b, and c ug/mL. etc.). This will help when it comes to comparing this study to the literature. In partucualr it is not clear how many different concentrations are tested as you only give the rang, you need to be more specific.
(Note that the CLSI guideline ref should be 24 according to your reference list, also I can't get that PDF from the link you provide, It is important to sort this reference out).
Author Response
Comments and Suggestions for Authors
In this study the presence of antimicrobial resistant E. coil up- and downstream of human habitation is investigated. Overall there is no significant difference between rates of resistance up- verses downstream, although interestingly there is more resistance to cephems upstream. There are changes in rates of resistance by season, with lower rates in winter.
This is a well conducted study and a well written paper. Results are clearly and concisely presented and I have no hesitation is recommending it for publication subject to a few minor points. Reporting of the statistics used is the most vital correction needed.
We would like to express our sincere gratitude for your evaluation of our research paper. Also, thank you for your polite and informative comments. We have been very helpful in your peer review in correcting the treatise.
Our responses and corrections are shown below.
Reply to comments and suggestions:
The revised points are shown in yellow highlight in the revised manuscript.
General points:
- Generally writing is clear and concise, and everything is understandable. However I have noted a few errors and a general read-though to catch any errors would be worthwhile.
=>Thank you for pointing out. We corrected the pointed-out parts and rechecked through the whole paper.
- I am not keen on the title, in particular the use of "natural river". All rivers are natural, otherwise they would be a canal. Please drop the use of "natural river" thought-out the paper. The title (and study) should reflect that is it upstream and downstream of human settlement/activity. Therefore you have pristine river upstream, and human impacted river downstream.
=>Your suggestion is justified. Throughout the paper, all expressions of “natural river” have been deleted and corrected. The title has been revised with reference to your suggestions.
2.1 Related, but a better definition of the difference between the sites, (e.g. human impacted or whatever) needs to be included in the text, e.g. line 13 but also throughout.
=> Through the papers (including Figures and Tables), the points where it is better to show the differences between the upstream and downstream sites have been corrected to be clearly distinguished the sites as your comment.
- EX) “pristine upstream” and “human-impacted downstream”.
- Statistical methods are not reported in the methods, but there are some P-values. It is not clear how statistics where calculated, as there is just one set of samples for each location and each time of year, so n=1. This needs to be cleared up before publication. Also, when reporting stats include degrees freedom, test-statistic, and effect size/coefficient.
=>As another reviewer pointed this out as well, we have changed our strategy of statistical test; we have simply compared and discussed relative difference between up- and down-stream samples without statistical test in the revised manuscript. All descriptions of statistical analysis such as p values and significant have been deleted in the text.
3.1 on a related note, be carful saying something is "significant" when no statistical test was carried out, e.g. remove "significant" lines 123, 161, and elsewhere, unless a statistical test was carried out.
=> We have removed "significant" in the text.
- Due to the journal putting the methods at the end, figure 1 (map) is out of place and needs to be moved or renamed.
=>Figure 1 (map) has been inserted into “Introduction” so that the reader can first understand the purpose of this paper and the state of the research site.
4.1 on the map, it would be good to put a pin showing the location of Miyazaki.
=>Revised Figure 1 has been revised to show the location of Miyazaki on the map, as our suggestion.
Specific points:
Abstract.
Line 12-13, could be more simple and clear. e.g. This study compared the antibiotic resistance profile of Escherichia coli upstream and downstream of human habitation.
=>The sentence has been corrected with reference to your indication. [line 13]
Line 15, antibiotics cant have antibiotic resistance. I think you mean
Resistance to one or more antibiotics...….
=>We have revised the sentence as you instructed. [line 14]
Introduction:
Line 38, "selected" is the wrong word.
=>We have revised the word. [line 36]
Line 44, reported daily? is there evidence of a paper per day? Regularly might be a better word.
=>We have revised the sentence as you instructed. [line 42]
Line 66, "away from the humans" is clumsy. Away from human activity would be better.
=>We have revised the sentence as you instructed. [line 63]
Results:
Table 1. are these values means? what is the standard deviation, how did you did statistics if no standard deviation or variance? please put the standard deviation on the plots. Would be even better to put a star or something to indicate values that are significantly different up- and downstream. (remember to detail stats in methods).
=>We evaluated the differences between the three surveys of upstream and downstream. However, due to the small number of data and the p-value of the t-test being greater than 0.05, the discussions of the significant difference test by statistical analysis have been deleted. As your suggestion, Table 1 has been modified.
Table 2. again you have stats, but not clear how as you don't have standard deviation?
=>The data are the percentage of the strain’s ratio. It cannot be expressed in SD. Table 2 has been modified to show total results instead of averages.
Line 148, be consistent with numbers "one to three" or "1 to 3",
=>We have revised the point as you instructed. [line 148]
Table 3, is cephemes spelt right? I think it is cephem?
=>We have revised the point as you instructed. [Table 3]
Methods:
Please provide more detail for the MIC testing. Unfortunately there is great variation in how the CLSI guidelines are interpreted, especially for environmental samples. Make it clear exactly what concentrations where used for each antibiotic, rather than the range (e.g. ampicillin at x, y, and z ug/mL, cegazolin at a, b, and c ug/mL. etc.). This will help when it comes to comparing this study to the literature. In partucualr it is not clear how many different concentrations are tested as you only give the rang, you need to be more specific.
=>As you pointed out, there are several MIC tests recommended by CLSI guideline, the agar dilution method was selected in this study. Since this method has advantage that different concentrations of antibiotics can be tested for many strains, we think it is suitable for the evaluation of MIC isolated from environmental strains. We have revised to make clear your suggested point in the section “3.4. Determination of minimum inhibitory concentration (MIC)”. [Line 280–297]
(Note that the CLSI guideline ref should be 24 according to your reference list, also I can't get that PDF from the link you provide, It is important to sort this reference out).
=>The available online sites were gone.
We have corrected the reference lists. The No. 24 in the references was error. No. 23 is correct. In addition, new reference 24 has been added to exact information. [Line 402–407]
Reviewer 2 Report
This is an interesting report on antimicrobial-resistant enterobacteria isolated from a river in Japan. The most interesting conclusion is that the number of antimicrobial-resistant bacteria is high, both at locations deep in the forest or close to human settlements. Overall, the manuscript is clear and well written, and the observations reported here are quite surprising with regards to the number of antimicrobial-resistant enterobacteria located in the upstream sampling region. It is also interesting to know that there are seasonal variations in the type of antimicrobial-resistant enterobacteria present in the river. However, some of the analyses and their interpretations are very speculative. At this point of the revision process, I have three major concerns:
- The upstream sampling location is very closely located to a wasteland according to Figure 1. Could this have an impact on the conclusions drawn by the authors?
- The statistical analysis of the report is based on p-values below or above 0.1. This is highly unusual, authors should apply a more stringent p-value, at least 0.05. I am therefore wondering if the results are statistically significant or not.
- Finally, the authors are suggesting that the high number of antimicrobial-resistant enterobacteria is due to the presence of seagulls or other wild animals. Is there any other evidence that may be drawn from the PFGE analysis reported here with regards to the most probable source of bacteria in each location? In particular, it would be interesting to test if some of the genotypes identified in Figure 3 could be associated with birds or other wild animals. On the other hand, is it possible that the source of enterobacteria isolated in the upstream location are domestic animals? E.g. free-ranging cattle or dogs accompanying humans walking in the area close to the river?
Minor comment: the species names in the bibliography are not written in italics.
Author Response
Comments and Suggestions for Authors
This is an interesting report on antimicrobial-resistant enterobacteria isolated from a river in Japan. The most interesting conclusion is that the number of antimicrobial-resistant bacteria is high, both at locations deep in the forest or close to human settlements. Overall, the manuscript is clear and well written, and the observations reported here are quite surprising with regards to the number of antimicrobial-resistant enterobacteria located in the upstream sampling region. It is also interesting to know that there are seasonal variations in the type of antimicrobial-resistant enterobacteria present in the river. However, some of the analyses and their interpretations are very speculative. At this point of the revision process, I have three major concerns:
We would like to express our sincere gratitude for your evaluation of our research paper.
Our responses and corrections to your concerns are shown below.
Reply to comments and suggestions:
The revised points are shown in yellow highlight in the revised manuscript.
The upstream sampling location is very closely located to a wasteland according to Figure 1. Could this have an impact on the conclusions drawn by the authors?
=>Since it is the wasteland area due to typhoons, there is no human activity. Therefore, it does not affect to this paper. The sentence has been added in as follows;
“Since the wasteland near the upstream site is a deforested land due to typhoon events, there is no human activity.” [Line 236–237]
The statistical analysis of the report is based on p-values below or above 0.1. This is highly unusual, authors should apply a more stringent p-value, at least 0.05. I am therefore wondering if the results are statistically significant or not.
=>We agree your suggestion. As responded to you and another reviewer, we have changed our strategy of statistical test; we have simply compared and discussed relative difference between up- and down-stream samples without statistical test in the revised manuscript.
Finally, the authors are suggesting that the high number of antimicrobial-resistant enterobacteria is due to the presence of seagulls or other wild animals. Is there any other evidence that may be drawn from the PFGE analysis reported here with regards to the most probable source of bacteria in each location? In particular, it would be interesting to test if some of the genotypes identified in Figure 3 could be associated with birds or other wild animals. On the other hand, is it possible that the source of enterobacteria isolated in the upstream location are domestic animals? E.g. free-ranging cattle or dogs accompanying humans walking in the area close to the river?
=>Thank you for your helpful comments. We are similarly interested in what you have pointed out. Currently, we are investigating and researching fecal bacteria in the upstream river water and in feces of birds and mammals that utilize the waterside of the upstream. We will report the result when it comes out.
Minor comment: the species names in the bibliography are not written in italics.
=>We have revised the references list.
Round 2
Reviewer 2 Report
I do not have any further comments.